# Infant Age Moderates Associations between Infant Temperament and Maternal Technology Use during Infant Feeding and Care

**DOI:** 10.3390/ijerph191912858

**Published:** 2022-10-07

**Authors:** Maya I. Davis, Camille M. Delfosse, Alison K. Ventura

**Affiliations:** Department of Kinesiology and Public Health, Center for Health Research, California Polytechnic State University, San Luis Obispo, CA 93407, USA

**Keywords:** maternal technology use, mobile device use, infant feeding, infant temperament, age

## Abstract

Previous research illustrated that infants’ temperamental traits shape parents’ behaviors, but parents’ behaviors can also elicit or intensify infants’ behaviors in ways that shape temperament. One understudied aspect of parenting that may exhibit bidirectional influences with temperament is parent technology use (e.g., use of mobile devices) within family contexts. To date, few studies have examined whether maternal technology use is associated with infant temperament and whether age-related differences in these associations exist. The present study was a secondary analysis of pooled data from three infant feeding studies. Mothers (*n* = 374) of young infants (age 16.2 ± 6.2 weeks) completed measures of maternal technology use during infant feeding and care interactions, infant temperament, and family demographics. Maternal technology use was positively associated with negative affectivity and negatively associated with orienting/regulatory capacity but was not associated with positive affectivity/surgency. The association between maternal technology use and negative affectivity was stronger for younger infants than older infants, while the association between maternal technology use and orienting/regulatory capacity was not significant for younger infants but was for older infants. Findings suggest maternal technology use is associated with infant negative affectivity and orienting/regulatory capacity, but the strength of these associations may change with infant age. Further longitudinal research is needed to verify this interpretation and understand mechanisms underlying these associations.

## 1. Introduction

Digital technologies, such as mobile phones and smartphones, have become a fixture in the lives of today’s families. Recent global data illustrate that 94% of adults in nations with advanced economies and 83% of adults in nations with emerging economies own a mobile phone or smartphone [1]. In addition, a recent study of US adults reported that 44% of 18- to 49-year-olds are online “almost constantly” [2]. The ubiquity of digital technologies and the ease with which they can be used have led to increased concern that technology use may impact our day-to-day activities and social interactions.

Technology use within families is of particular concern. Parents report spending an average of 5.5 h per day using technology and mobile devices [3], and 68% of parents report being distracted by their smartphones when spending time with their children [4]. Parents of young infants may be particularly likely to use technology during infant feeding and care. Indeed, in a recent study, 43% of mothers reported engaging with distractions during feeding and 26% reported engaging with technological distractions [5]. In a different study, over one-third of mothers surveyed disclosed they always or often engaged with technology during infant feeding and care interactions, with 40% reporting they always or often watched TV and 37% always or often used a mobile device or tablet [6]. Thus, this emerging evidence suggests that technology use during infant feeding and care interactions may be a common experience for mothers and infants, but further research is needed to understand the potential implications of these trends for mother–infant interactions and infant behavior and development.

On one hand, technology use likely brings important benefits to parents and may be particularly appealing to parents of young children as a way to escape boredom or stress that may come with parenting [7]. Previous research illustrated that more frequent maternal technology use is associated with greater levels of infant temperamental negative affectivity, characterized by higher levels of crying, fearfulness, and anger, and lower levels of soothability and stimulation threshold [8]. This association may represent mothers’ reactions to their parenting experiences and use of technology to cope with a more challenging infant temperament. On the other hand, mothers’ technology use may detract from mother–child interactions, as suggested by the displacement hypothesis, which posits time spent using technological devices decreases important family connections [9]. In support of this hypothesis, one study found that 96% of mothers perceived that mobile devices interfered with their family’s interactions [10]. Additionally, experimental studies illustrated that, when mothers withdraw from mother–infant interactions to attend to their mobile devices, their infants exhibit decreases in positive affect and increases in negative affect and bids for attention [11,12,13]. Parenting behaviors that elicit or intensify certain infant behaviors, such as positive and negative affectivity, can shape infant temperament and related socioemotional outcomes [14]. Thus, repeated displacement of mother–infant interaction with maternal technology use paired with negative impacts of this displacement on infant behavior may lead to associations between maternal technology use and infant temperament that strengthen over time. Taken together, it is possible that bidirectional associations between mothers’ technology use and infant temperament exist, as mothers may use technology to cope with infant behaviors they perceive to be difficult, but tendencies to use technology to cope may also exacerbate difficult infant behaviors and shape temperamental characteristics across infancy [14,15].

In sum, recent research examined the potential benefits versus detriments of parent technology use within family contexts, as well as how this relates to child characteristics, such as temperament. Emerging research suggests maternal technology use is associated with infant behavior and temperament, but age-related differences in these associations may exist. The aim of the present study was to further examine associations between maternal technology use and dimensions of infant temperament (i.e., negative affectivity, orienting/regulatory capacity, and positive affectivity/surgency) and to explore whether these associations differ for younger versus older infants. On the basis of previous research, we hypothesized that greater maternal technology use would be associated with greater temperamental negative affectivity and lower orienting/regulatory capacity and positive affectivity/surgency. We also hypothesized that infant age would modify these associations in that associations would be stronger for older versus younger infants.

## 2. Materials and Methods

### 2.1. Participants

The present study was a secondary analysis of pooled data (*n* = 374) from three previous infant feeding studies (for more details, see [6,16,17]). The first two studies took place in laboratory sites in San Luis Obispo, CA, and Philadelphia, PA, but mothers completed all surveys within an online platform. The final study was an online survey and did not include an in-person assessment. Inclusion criteria for these studies were mothers aged 18 years and older, and infants who were singletons born at a gestational age of at least 37 weeks (full-term) and were between 2–6 months of age at the time of the study. Dyads were excluded if infants had any medical conditions or developmental delays that might interfere with their feeding. In two of the studies, mothers were recruited through ads in local newspapers, nutrition assistance program offices, fliers in libraries, coffee shops, and local pediatric offices, local parent support groups, and online sites (e.g., Facebook). The last study used Amazon Mechanical Turk (MTurk), a crowdsourcing platform to recruit participants, who, if they met the study criteria, were then directed to a Qualtrics survey. All study procedures were reviewed and approved by the Drexel University and/or California Polytechnic State University Institutional Review Boards. 

### 2.2. Measures

#### 2.2.1. Demographics Questionnaire

Family demographic information was assessed using a questionnaire developed by the research team. The questionnaires assessed mothers’ parity, age, education level, race/ethnicity, family income, marital and employment status, and participation in federal nutrition assistance programs.

#### 2.2.2. Infant Behavior Questionnaire—Very Short Form

The Infant Behavior Questionnaire—Revised Very Short Form (IBQ-RVS) [18,19] was used to assess infant temperament on the basis of mothers’ perceptions of infant reactivity and self-regulatory behaviors that are phenotypes of temperament. The current study used a version of the standard IBQ, abbreviated from 184 items to 37, to reduce participant burden. Each of the 37 questions represents 16 scales, further divided into three subscales of infant temperament: negative affectivity, orienting/regulatory capacity, and positive affectivity/surgency. Negative affectivity is represented by sadness, distress to limitations, fear, and falling reactivity (example item: “When you were busy with another activity, and your baby was not able to get your attention, how often did s/he cry?”). Orienting/regulatory capacity is represented by low-intensity pleasure, cuddliness/affiliation, duration of orienting, and soothability (example item: “How often during the last week did the baby play with one toy or object for 5–10 min?”). Positive affectivity/surgency is represented by approach, vocal reactivity, high intensity pleasure, smiling and laughter, activity level, and perceptual sensitivity (example item: “How often during the last week did your baby smile or laugh when given a toy?”). Each item was scored on a Likert scale of 0 (never) to 7 (always). The IBQ-RVS has been validated in diverse populations of mothers of infants younger than 3 months to up to 3 years old, with subscales demonstrating good internal consistency: negative affectivity (α = 0.78), orienting/regulatory capacity (α = 0.75), and positive affectivity/surgency (α = 0.77) [19]. These subscales also demonstrated good internal consistency in the present study: negative affectivity (α = 0.84), orienting/regulatory capacity (α = 0.75), and positive affectivity/surgency (α = 0.84).

#### 2.2.3. Maternal Distraction Questionnaire

The Maternal Distraction Questionnaire (MDQ) is a validated self-report measure of the various activities that mothers may engage in while interacting with their infants, during both feeding interactions and nonfeeding (e.g., soothing and play) interactions. Development of MDQ items was based, in part, on the research team’s previous qualitative work in which mothers were asked to keep feeding diaries [5,20]. Diaries were kept for three consecutive days, during which the mother recorded the following each time they fed their infants: (1) the feeding start and end time; (2) what was fed (e.g., formula, breast milk from the breast, breast milk from a bottle); (3) the amount fed (if known); (4) what else, if anything, they were doing while feeding their infants. Analysis of the last question revealed that mothers most commonly reported doing the following activities while feeding their infants: (1) watching television; (2) using a smartphone or tablet; (3) using a computer; (4) talking on the phone or to another adult; (5) sleeping; (6) reading a book, magazine, or newspaper [5,20]. While developing the MDQ, the research team reviewed published measures assessing distraction in different settings (e.g., multitasking [21]) to determine questionnaire structure and other activities that should be included. The result was the MDQ, an 18-item questionnaire in which the mother is asked a series of questions related to how often she engages in common activities (e.g., watching television, talking or texting on the phone, using the computer, or reading a magazine) during infant feeding interactions and other infant care interactions. For each item, the mother used a five-point Likert scale comprising 1 (never), 2 (rarely), 3 (sometimes), 4 (often), and 5 (always). The MDQ has been validated in a sample of mothers of infants under 12 months of age, with the maternal technology use during infant feeding and care (referred to hereafter as maternal tech use) subscale demonstrating good internal consistency (α = 0.86) [6]. The maternal tech use subscale also demonstrated good internal consistency in the present study (α = 0.84).

### 2.3. Statistical Analysis

All statistical analyses were conducted using SAS v.9.4 (SAS Institute Inc., Cary, NC, USA). Prior to statistical analyses, data were cleaned and assessed for normality. Descriptive statistics were calculated to describe sample characteristics. General linear regression models (six in total) were used to examine associations between frequency of maternal tech use and dimensions of infant temperament (negative affectivity, orienting/regulatory capacity, and positive affectivity/surgency) and whether infant age moderated associations between maternal tech use and dimensions of infant temperament (tested by adding an infant age by maternal tech use interaction to all models). All models were controlled for maternal age, education level, marital status, parity, family income level, and race/ethnicity. A threshold of *p* < 0.05 was used for statistical significance.

## 3. Results

### 3.1. Sample Characteristics

Sample characteristics are reported in Table 1. Infants were on average 16.2 weeks (SD 6.2, range: 2.2–31 weeks), and slightly over half (52%) were female. Mothers were on average 31.5 years of age (SD 4.5). Thirty-four percent of mothers were primiparous, 33% were enrolled in federal nutrition assistance programs, 46% had a college degree, and 75% were non-Hispanic White.

### 3.2. Associations between Maternal Tech Use and Infant Temperament

As illustrated in Table 2, frequency of maternal tech use was positively associated with negative affectivity (β = 0.40, SE = 0.07, *p* < 0.001) and negatively associated with orienting/regulatory capacity (β = −0.13, SE = 0.06, *p* = 0.02), but was not associated with positive affectivity/surgency (β = 0.04, SE = 0.08, *p* = 0.62). Thus, greater frequency of maternal tech use was associated with greater negative affectivity and lower orienting/regulatory capacity but was not significantly associated with positive affectivity/surgency.

### 3.3. Infant Age Moderates Associations between Maternal Tech Use and Infant Temperament

As illustrated in Table 3, infant age moderated associations between maternal tech use and negative affectivity (*p* = 0.01) and orienting/regulatory capacity (*p* = 0.04), but the interaction between infant age and maternal tech use within the model predicting positive affectivity/surgency did not reach significance (*p* = 0.05). The positive association between increased frequency of maternal tech use and infant negative affectivity was significantly stronger for younger infants (age −1 SD below average; *p* < 0.001) than for older infants (age +1 SD above average; *p* = 0.03) (Figure 1). The negative association between increased frequency of maternal tech use and infant orienting/regulatory capacity was not significant for younger infants (*p* = 0.94) but was significant for older infants (*p* = 0.01; Figure 2).

## 4. Discussion

Many parents report that their use of smartphones and other technological devices interferes with family interactions [4]. This trend is concerning given parents’ active attention to and coregulation with their young children during early interactions help to shape children’s cognitive development, psychomotor abilities, and secure attachments [22]. However, more research is needed to understand associations between parent technology use and markers of infant behavior and development, such as infant temperament, and how these associations may change over time. 

The present study was an initial attempt to address this research gap through a secondary analysis of data from infant feeding studies. Findings from the present study illustrated that more frequent maternal technology use during infant feeding and care was associated with greater maternal-reported infant negative affectivity and lower orienting/regulatory capacity, but these associations were modified by infant age. More frequent maternal technology use was associated with greater levels of negative affectivity in younger infants; this association was present for older infants but was weaker. However, the association between maternal technology use and orienting/regulatory capacity was not significant for younger infants, but more frequent maternal technology use was associated with lower orienting/regulatory capacity for older infants. 

The first year postpartum may be a time when mothers are particularly apt to engage with technological distractors such as television and mobile devices [5,6,20,23]. During early infancy, young infants feed 8–12 times per day and these feedings can last anywhere between 5 and 60 (or more) min. In addition, time spent feeding and caring for infants is typically also the time when mothers’ abilities to multitask or leave the home are limited. Thus, it is quite plausible that engagement with distracting technology is an especially attractive way for mothers of young infants to cope with the boredom or frustration that might come with the large volume of time they must devote to feeding or caring for their infants. In many ways, integration of technology use within family life makes sense and likely provides many important benefits to mothers and families that should be recognized and preserved [3,7,24,25].

However, it is also widely recognized that absorption in technology and mobile devices creates a state of “present absence” wherein a mother may be physically present with her child, but attentionally and emotionally absent [12,13,24]. Recent experimental studies illustrate that, when mothers withdraw from mother–infant interactions to engage with mobile devices, infants notice and react negatively by increasing their expression of negative affect and decreasing their expression of positive affect and orienting toward their mothers [12,13]. Thus, when interpreted within the context of what is known from these previous experimental studies, the associations between maternal technology use, negative affectivity, and orienting/regulatory capacity noted in the present study may represent a phenomenon wherein the present absence created when mothers repeatedly disengage from mother–infant interactions to attend to mobile devices negatively impacts infant behavior in ways that lead to greater negative affectivity and lower orienting/regulatory capacity over time.

However, novel findings of the present study were that infant age moderated associations between maternal technology use and dimensions of infant temperament. The cross-sectional nature of these data precludes the determination of causal effects, but there are several possible interpretations of these interactions that could be verified with further longitudinal and experimental research. With respect to the age difference in the strength of the positive association between maternal technology use and infant negative affectivity, younger infants may be more affected by maternal technology use than older infants, leading to more crying, distress, and related behaviors that characterize negative affectivity. This interpretation is consistent with previous research with slightly older infants, which illustrated that more frequent maternal technology use during infant feeding and care interactions were associated with a greater display of distress-related behaviors in response to mothers’ mobile device use for younger (5–9 months old) but not older (10–14 months old) infants [13]. In a different study, infants were less distracted by the presence of technology during an observed feeding interaction if their mothers typically used technology during infant feeding [17]. Taken together, these findings support the possibility that infants habituate to maternal technology use over time, exhibiting less reactivity to mothers’ use of technology when it is an expected presence during mother–infant interactions.

With respect to the age difference in the significance of the negative association between maternal technology use and infant orienting/regulatory capacity, our finding that maternal technology use was associated with poorer orienting/regulatory capacity for older but not younger infants aligns with previous research also illustrating that more frequent maternal technology use was associated with poorer abilities to regulate negative emotions for older (10–14 months of age) but not younger (5–9 months of age) infants [13]. These significant, negative associations between maternal technology use and orienting/regulatory capacities seen for older, but not younger, infants may represent effects of maternal technology use on the development of self-regulation over time. Previous cross-sectional and observational studies suggested that maternal technology use is associated with lower maternal sensitive responsiveness to young children’s cues and bids for attention [17,20,26,27,28,29], which is an important support for the development of self-regulatory skills during the first year. Thus, repeated experiences of lower maternal sensitivity and responsiveness to infant cues mediated by maternal technology use may disrupt the scaffolded support infants need from mothers to learn how to self-regulate behavior and emotions.

However, given the cross-sectional nature of the present study and the reliance on maternal reports of technology use and infant temperament, it is important to acknowledge the equal plausibility of alternative explanations for study findings that do not imply causal impacts of maternal technology use on infant behavior and development. Mothers’ perceptions of and experiences with their infants likely influence their propensity to use technology in the presence of their infants. Thus, an alternative interpretation of study findings is that mothers of younger versus older infants react differently to infant behaviors representing greater negative affectivity (e.g., greater expressions of sadness, distress to limitations, fear, and falling reactivity) or lower orienting/regulatory capacities (e.g., lower levels of low intensity pleasure, cuddliness, orienting, and soothability). Given that infant temperament was assessed by maternal report (and, thus, represented mothers’ perceptions of infant behaviors that are phenotypes of temperament), the present findings may represent effects of maternal technology use on mothers’ perceptions and interpretations of infant behavior, or vice versa. Indeed, certain mothers may also be more apt to use technology to cope with the stresses of parenting, especially when they find their infants’ behavior to be more challenging. It is also possible that unmeasured variables, such as maternal depression, influence both maternal technology use and perceptions of infant behavior. Thus, further longitudinal research is needed to verify the above interpretations and to better understand potential bidirectional influences between maternal technology use and infant temperament over time.

It is also important to note additional limitations of the present study that may be addressed by future studies. The present study explored age differences; while informative, these differences may not represent age-related changes over time. Further longitudinal and experimental research is needed to further understand the potential causal mechanisms and developmental trends underlying noted associations. Additionally, online sites were used for both recruitment and data collection. Thus, our sample may have been more technologically savvy than a sample recruited and assessed using nontechnological methods. Furthermore, as indicated above, all data were self-reported by participants, and some reported high levels of technological and other distractions during infant feeding and care, increasing susceptibility to reporting bias. Further research with larger, more diverse samples and objective measures of mothers’ technology use and infant behavior and development is warranted.

## 5. Conclusions

Technology use provides many benefits to parents and families, and these benefits may be especially important for mothers of infants. However, emerging research suggests that bidirectional influences exist between maternal technology use and infant behavior and temperament; the frequency with which mothers use technology during infant feeding and care interactions may impact mother–infant interactions and infant behavior and development, but infants’ temperamental traits may also impact mothers’ willingness and perceived need to engage with technology. The present study examined this issue by exploring associations between maternal technology use and infant temperament, as well as whether age-related differences in these associations existed. Findings suggest maternal technology use is associated with infant negative affectivity and orienting/regulatory capacity, but the strength of these associations may change with infant age. Additional longitudinal research is needed to verify these conclusions and address research gaps related to understanding longitudinal associations between maternal technology use and infant behavior and developmental outcomes.

## Figures and Tables

**Figure 1 ijerph-19-12858-f001:**
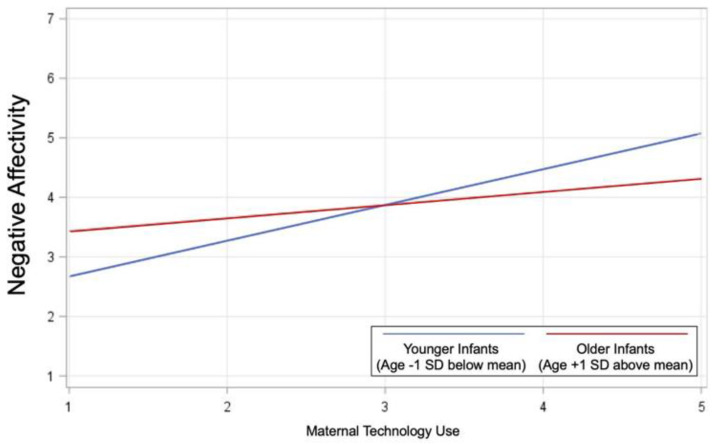
Infant age moderated associations between maternal tech use and negative affectivity (*p* = 0.01). The positive association between increased frequency of maternal tech use and infant negative affectivity was significantly stronger for younger infants (age −1 SD below average; *p* < 0.001) than for older infants (age +1 SD above average; *p* = 0.03).

**Figure 2 ijerph-19-12858-f002:**
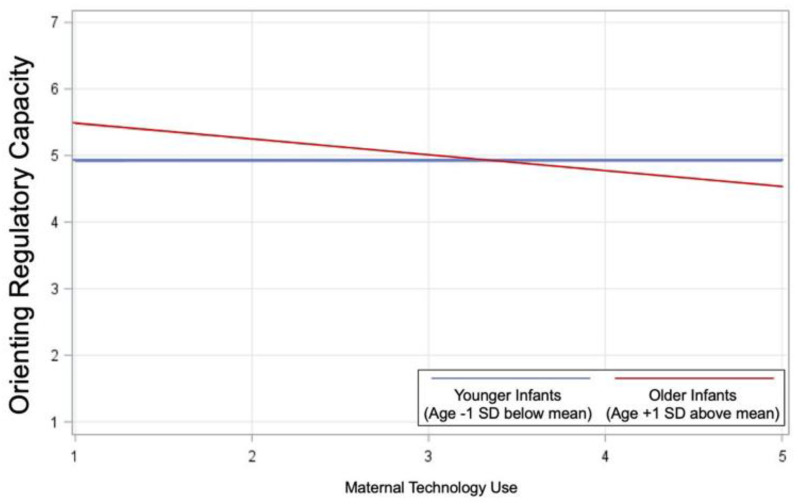
Infant age moderated associations between maternal tech use and orienting/regulatory capacity (*p* = 0.04). The negative association between increased frequency of maternal tech use and infant orienting/regulatory capacity was not significant for younger infants (age −1 SD below average; *p* = 0.94) but was significant for older infants (age +1 SD above average; *p* = 0.01).

**Table 1 ijerph-19-12858-t001:** Sample characteristics (*n* = 374).

Characteristic	% (*n*) or Mean (SD)
*Infants*	
Age in weeks, mean (SD)	16.2 (6.2)
Sex, % (*n*) female	52.4 (196)
*Mothers*	
Age in years, mean (SD)	31.5 (4.5)
Education level	
Did not complete high school	1.6 (6)
High-school degree	9.6 (36)
Some college or vocational degree	42.5 (159)
Bachelors or graduate degree	46.3 (173)
Marital status, % (*n*) married	70.8 (264)
Parity, % (*n*) primiparous	34.1 (127)
Annual family income level	
<15,000 USD/year	5.6 (21)
15,000–<35,000 USD/year	21.4 (80)
35,000–<75,000 USD/year	41.2 (154)
>75,000 USD/year	30.2 (113)
Not reported	1.6 (6)
Race/ethnicity	
Non-Hispanic, White	75.1 (281)
Non-Hispanic, Black	7.5 (28)
Hispanic, any race	10.4 (39)
Asian	6.1 (23)
Not reported	0.8 (3)

**Table 2 ijerph-19-12858-t002:** Associations between maternal technology use during infant feeding and care and dimensions of infant temperament.

	Model 1			Model 2			Model 3		
	Negative Affectivity			Orienting/Regulatory Capacity			Positive Affectivity/ Surgency		
	Estimate	SE	*p*-Value	Estimate	SE	*p*-Value	Estimate	SE	*p*-Value
Intercept	3.19	0.49	<0.001	5.22	0.39	<0.001	3.64	0.51	<0.001
Mother’s age	−0.02	0.01	0.17	0.00	0.01	0.93	−0.02	0.01	0.14
Mother’s education level									
Did not complete high school	−0.19	0.47	0.68	−0.31	0.37	0.40	−0.45	0.49	0.36
High school degree	0.13	0.18	0.47	−0.04	0.14	0.78	0.00	0.19	0.99
Some college or vocational degree	0.12	0.11	0.26	−0.07	0.09	0.41	0.06	0.11	0.62
Bachelors or graduate degree	Ref	-	-	Ref	-	-	Ref	-	-
Marital status									
Not married	−0.23	0.11	0.05	0.09	0.09	0.32	0.15	0.12	0.21
Married	Ref	-	-	Ref	-	-	Ref	-	-
Parity, % (*n*) primiparous									
Multiparous	0.04	0.12	0.72	−0.05	0.08	0.52	0.05	0.11	0.64
Primiparous	Ref	-	-	Ref	-	-	Ref	-	-
Annual family income level									
<15,000 USD/year	0.26	0.24	0.27	0.12	0.19	0.54	−0.09	0.25	0.72
15,000–<35,000 USD/year	−0.01	0.15	0.99	0.10	0.12	0.40	0.02	0.15	0.88
35,000–<75,000 USD/year	0.20	0.12	0.27	0.12	0.10	0.27	0.25	0.13	0.04
>75,000 USD/year	Ref	-	-	Ref	-	-	Ref	-	-
Race/ethnicity									
Non-Hispanic, White	Ref	-	-	Ref	-	-	Ref	-	-
Non-Hispanic, Black	0.24	0.20	0.20	0.16	0.15	0.30	0.16	0.20	0.41
Hispanic, any race	0.01	0.16	0.95	0.24	0.13	0.06	0.39	0.17	0.02
Asian	0.34	0.20	0.08	−0.34	0.16	0.03	−0.24	0.21	0.25
Infant age	0.01	0.01	0.35	0.01	0.01	0.06	0.07	0.01	<0.001
Maternal technology use	0.40	0.07	<0.001	−0.13	0.06	0.02	0.04	0.08	0.62

**Table 3 ijerph-19-12858-t003:** Infant age moderates associations between maternal technology use and dimensions of infant temperament.

	Model 1			Model 2			Model 3		
	Negative Affectivity			Orienting/Regulatory Capacity			Positive Affectivity/Surgency		
	Estimate	SE	*p*-Value	Estimate	SE	*p*-Value	Estimate	SE	*p*-Value
Intercept	1.81	0.71	0.01	4.36	0.56	<0.001	2.56	0.74	0.01
Mother’s age	−0.02	0.01	0.14	0.00	0.01	0.86	−0.02	0.01	0.12
Mother’s education level									
Did not complete high school	−0.14	0.46	0.77	−0.27	0.37	0.46	−0.40	0.48	0.41
High school degree	0.11	0.18	0.54	−0.05	0.14	0.71	−0.01	0.18	0.94
Some college or vocational degree	0.16	0.11	0.15	−0.05	0.09	0.56	0.08	0.11	0.47
Bachelors or graduate degree	Ref	-	-	Ref	-	-	Ref	-	-
Marital status									
Not married	−0.22	0.12	0.06	0.10	0.09	0.27	0.16	0.12	0.18
Married	Ref	-	-	Ref	-	-	Ref	-	-
Parity, % (*n*) primiparous									
Multiparous	0.04	0.11	0.70	−0.05	0.08	0.53	0.05	0.11	0.63
Primiparous	Ref	-	-	Ref	-	-	Ref	-	-
Annual family income level									
<15,000 USD/year	0.26	0.24	0.28	0.11	0.19	0.56	−0.10	0.25	0.70
15,000–<35,000 USD/year	−0.04	0.15	0.79	0.08	0.12	0.50	−0.00	0.15	0.98
35,000–<75,000 USD/year	0.18	0.12	0.13	0.10	0.09	0.32	0.24	0.13	0.06
>75,000 USD/year	Ref	-	-	Ref	-	-	Ref	-	-
Race/Ethnicity									
Non-Hispanic, White	Ref	-	-	Ref	-	-	Ref	-	-
Non-Hispanic, Black	0.21	0.19	0.26	0.14	0.15	0.36	0.14	0.20	0.48
Hispanic, any race	0.01	0.16	0.97	0.24	0.13	0.06	0.39	0.17	0.02
Asian	0.35	0.20	0.08	−0.33	0.16	0.03	−0.23	0.20	0.25
Infant age	0.09	0.03	0.01	0.06	0.03	0.01	0.13	0.03	0.001
Maternal technology use	0.91	0.20	<0.001	0.19	0.16	0.25	0.44	0.21	0.04
Infant age × maternal technology use	−0.03	0.01	0.01	−0.02	0.01	0.04	−0.02	0.01	0.05

## Data Availability

The data for this study are available upon request; the lead author has full access to the data reported in the manuscript.

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
