# Peer review of "Infant Age Moderates Associations between Infant Temperament and Maternal Technology Use during Infant Feeding and Care"

_ijerph, 2022, doi:10.3390/ijerph191912858_

Round 1
Reviewer 1 Report
IJERPH Review
Infant age moderates associations between infant temperament and maternal technology use during infant feeding and care.
Thank you so much for giving me the opportunity to read this manuscript.
This research theme was new and important in the modern era. The manuscript itself was well-written and comprehensible with the correct way of statistics. I really enjoyed reading this manuscript and looking forward to its publishment very soon.
Here are my minor comments.
1. Infant temperament
Temperament is regarded as an infant’s trait (something unchanged) generally. However, the infant temperaments were interpreted as an infant’s behavior (or something changeable ) according to the situation (state) in the manuscript. Authors need to clarify how they recognize ‘temperament’ in the introduction (state or trait).
Another point is infant temperament was assessed by mothers which is regarded as perceived temperament. In the discussion part, the authors may need to discuss this point for the interpretation of the results.
There may be another interpretation; A mother who is mentally stressed or addicted to mobile techs may be more likely to judge her child as difficult.
2. Conclusions
Sentences between L314 and L
Author Response
RESPONSE TO REVIEWER 1’S COMMENTS:
REVIEWER COMMENT: Thank you so much for giving me the opportunity to read this manuscript. This research theme was new and important in the modern era. The manuscript itself was well-written and comprehensible with the correct way of statistics. I really enjoyed reading this manuscript and looking forward to its publishment very soon. Here are my minor comments.
- Infant temperament
Temperament is regarded as an infant’s trait (something unchanged) generally. However, the infant temperaments were interpreted as an infant’s behavior (or something changeable) according to the situation (state) in the manuscript. Authors need to clarify how they recognize ‘temperament’ in the introduction (state or trait).
AUTHOR RESPONSE: We thank the reviewer for bringing up this important point. We have revised our introduction so we now more thoroughly discuss previous research that illustrates bidirectional influences between parenting and temperament –although there is research to support some stability in temperament, there is also a substantial body of work that illustrates that just as child behavior can elicit certain behavioral responses that shape parents and parenting behaviors, parenting behaviors can also elicit behavioral responses in children that shape dimensions of child temperament (e.g., negative affectivity, positive affectivity, orienting/regulatory capacity) over time (for a review, see Kiff CJ, Lengua LJ, Zalewski M. Nature and nurturing: parenting in the context of child temperament. Clin Child Fam Psychol Rev. 2011 Sep;14(3):251-301).
REVIEWER COMMENT: Another point is infant temperament was assessed by mothers which is regarded as perceived temperament. In the discussion part, the authors may need to discuss this point for the interpretation of the results.
There may be another interpretation; A mother who is mentally stressed or addicted to mobile techs may be more likely to judge her child as difficult.
AUTHOR RESPONSE: We agree with both of these points and have revised our manuscript and discussion to better acknowledge the self-report nature of our measure of infant temperament and the possibility that mothers’ tech use of stress levels may influence their perceptions and interpretations of their infants’ behaviors. In particular, please see lines 228, 293-312
REVIEWER COMMENT: 2. Conclusions
Sentences between L314 and L
AUTHOR RESPONSE: This comment appears to be cut off – we would be happy to consider further suggestions from the reviewer, if needed.
Reviewer 2 Report
The present study used pooled data from three studies to examine maternal use of technology during feeding/care, infant temperament and infant age. This is a highly original study and important for understanding the modern context of feeding and care. The consideration of infant temperament adds to the quality of the study by including consideration of the infant’s role in the interaction with technology.
The manuscript is well written. Explanations are clear and the writing is efficient.
I only have a few points. The most important is that research questions or hypotheses are needed. Even though little is known in this area, there must have been questions driving this research and it is important to make these explicit.
Keep in mind that this is an international journal. The first paragraph should provide an international perspective. Line 110 should use another term for ‘WIC’ as it may not be meaningful to many readers, annual family income could be interpreted e.g low, medium, high, rather than just the amount.
The sentence lines 75-77 doesn’t add anything. It repeats the information in the previous sentence.
Author Response
RESPONSE TO REVIEWER 2'S COMMENTS:
REVIEWER COMMENT: The present study used pooled data from three studies to examine maternal use of technology during feeding/care, infant temperament and infant age. This is a highly original study and important for understanding the modern context of feeding and care. The consideration of infant temperament adds to the quality of the study by including consideration of the infant’s role in the interaction with technology.
The manuscript is well written. Explanations are clear and the writing is efficient.
AUTHOR RESPONSE: Thank you for the positive evaluation!
REVIEWER COMMENT: I only have a few points. The most important is that research questions or hypotheses are needed. Even though little is known in this area, there must have been questions driving this research and it is important to make these explicit.
AUTHOR RESPONSE: We have added hypotheses to the end of the introduction. Please see lines 72-83
REVIEWER COMMENT: Keep in mind that this is an international journal. The first paragraph should provide an international perspective. Line 110 should use another term for ‘WIC’ as it may not be meaningful to many readers, annual family income could be interpreted e.g low, medium, high, rather than just the amount.
AUTHOR RESPONSE: Thank you for these reminders. The first paragraph has been revised to have an international perspective (see lines 28-34). We replaced “WIC program offices” with “nutrition assistance program offices” to make this more meaningful to international readers (see line 95 and 105, Table 1). However, we opted not to classify annual family income as low, medium and high as we feel this classification depends on additional factors such as family size and cost of living in different regions of the U.S.; we do not have sufficient information to accurately classify these income levels, thus feel most comfortable reporting the income level ranges currently presented in the tables.
REVIEWER COMMENT: The sentence lines 75-77 doesn’t add anything. It repeats the information in the previous sentence.
AUTHOR RESPONSE: We have revised this part of the introduction section accordingly. Please see lines 67-71.